# A Bioassay Using a Pentadecanal Derivative to Measure S1P Lyase Activity

**DOI:** 10.3390/ijms22031438

**Published:** 2021-02-01

**Authors:** Kyong-Oh Shin, Maftuna Shamshiddinova, Jung-No Lee, Kwang-Sik Lee, Yong-Moon Lee

**Affiliations:** 1College of Pharmacy, Chungbuk National University, Cheongju, Chungbuk 28644, Korea; 0194768809@hanmail.net (K.-O.S.); shamshiddinovamaftuna@gmail.com (M.S.); 2Bio Convergence R&D Center, CoSeedBioPharm Corporation, Heungdeok-gu, Cheongju, Chungbuk 28161, Korea; rnd@coseed.co.kr; 3Future Science Research Center, Coreana Cosmetics Corporation, Chungnam 31041, Korea; kslee@coreana.co.kr

**Keywords:** sphingosine 1-phosphate lyase, HPLC, fluorescence detection, 5,5-dimethyl cyclohexanedione, hexadecenal, pentadecanal

## Abstract

Sphingosine-1-phosphate (S1P) is a unique lipid ligand binding to S1P receptors to transduce various cell survival or proliferation signals via small G proteins. S1P lyase (S1PL) is the specific enzyme that degrades S1P to phosphoethanolamine and (2E)-hexadecenal and therefore regulates S1P levels. S1PL also degrades dihydrosphingosine-1-phosphate (Sa1P), with a higher affinity to produce hexadecanal. Here, we developed a newly designed assay using a C17-Sa1P substrate that degrades into pentadecanal and phosphoethanolamine. For higher sensitivity in pentadecanal analysis, we developed a quantitative protocol as well as a 5,5-dimethyl cyclohexanedione (5,5-dimethyl CHD) derivatization method. The derivatization conditions were optimized for the reaction time, temperature, and concentrations of the 5,5-dimethyl CHD reagent, acetic acid, and ammonium acetate. The S1PL reaction in the cell lysate after spiking 20 µM of C17-Sa1P for 20 min was linear to the total protein concentrations of 50 µg. The S1PL levels (4 pmol/mg/min) were readily detected in this HPLC with fluorescence detection (λex = 366 nm, λem = 455 nm). The S1PL-catalyzed reaction was linear over 30 min and yielded a K_m_ value of 2.68 μM for C17-Sa1P. This new method was validated to measure the S1PL activity of mouse embryonal carcinoma cell lines of the standard cell (F9-0), S1PL knockdown cells (F9-2), and S1PL-overexpressed cells (F9-4). Furthermore, we treated F9-4 cells with different S1PL inhibitors such as FTY720, 4-deoxypyridoxine (DOP), and the deletion of pyridoxal-5-phosphate (P5P), an essential cofactor for S1PL activity, and observed a significant decrease in pentadecanal relative to the untreated cells. In conclusion, we developed a highly sensitive S1PL assay using a C17-Sa1P substrate for pentadecanal quantification for application in the characterization of S1PL activity in vitro.

## 1. Introduction

Bioactive sphingosine 1-phosphate (S1P) regulates cellular processes such as cell proliferation, differentiation, migration, and cell death [1,2,3]. S1P signals through a family of five G-protein-coupled membrane receptors (GPCRs) that regulate complex physiological processes such as vascular maturation and lymphocyte circulation [4,5,6]. S1P levels in our body seem to be controlled by three specific enzymes: S1P synthesis by sphingosine kinases (type I and II) and S1P degradations by S1P lyase (S1PL) and S1P phosphatase [7,8]. S1PL is an ER membrane protein that catalyzes the irreversible cleavage of S1P at the C2–C3 carbon–carbon bond, resulting in the reduction of S1P and yielding (E)-2-hexadecenal and ethanolamine phosphate [9,10]. Specifically, S1PL also degrades dihydro-S1P (Sa1P), resulting in the formation of hexadecanal (Matloubian, 2004). The inhibition of S1PL by 4-deoxypyridoxine (DOP) or 2-acetyl-5-tetrahydroxybutyl imidazole (THI) previously produced lymphopenia in mice, which may be useful in the treatment of autoimmune diseases [11,12]. Therefore, S1PL has emerged as a novel therapeutic target in various immune-related disorders, and thus studies on the pharmaceutical modulation of S1PL activity could be useful in therapeutic contexts [13]. S1PL activity has been measured using radioactive, fluorescent substrates. In an initial study, the incubation of the enzyme with [4,5-3H] Sa1P or S133P-biotin yielded its radioactive aldehyde. It was separated by thin-layer chromatography and quantified by a radiometric facility [14,15]. However, radioactive substrate usage requires multiple cumbersome separation steps, such as phase separation and scraping the tritium-labeled reaction product from a thin-layer chromatographic plate. The fluorescent substrates, such as 7-nitrobenz-2-oxa-1,3-diazol S1P (NBD-S1P), were employed to enhance the detectability by fluorescently monitoring S1PL products [16]. However, fluorescent substrates, the appearance of multiple fluorescent byproducts, or a higher K_m_ value are the main disadvantages of each method. Recently, to overcome these disadvantages, a specific chemical method for targeting aldehyde products was introduced, producing intense mass fragmentation ions such as semicarbazone [17], 2-diphenylacetyl-1,3-indandione-1-hydrazone (DAIH) derivatives, and biotinylated aminosphingosine-1-[(33)P]phosphate (S1(33)P-biotin) [15,18]. However, the intense mass fragment ions originating from the reagent itself provide high sensitivity in the MRM mode, and nonspecific reactions with other functional groups in biological samples sometimes exhibit complex chromatograms, disturbing the detection of corresponding aldehydes. Therefore, the ideal designed reagent is nonfluorescent and reacts with an aldehyde to selectively form a fluorogenic structure.

Here, we introduce a new method applying 5,5-dimethyl cyclohexane-1,3-dione (5,5-dimethyl cyclohexanedione (5,5-dimethyl CHD)) derivatization for selectively detecting an aldehyde product with S1PL. To avoid possible interference by endogenous fatty aldehydes, we used C17-dihydrosphingosine 1-phosphate (C17-Sa1P) as the substrate disintegrated into phosphoethanolamine and pentadecanal with S1PL. Heptadecanal as an internal standard with an odd number of carbon atoms and separated well from pentadecanal on a chromatogram was used. The HPLC-equipped fluorescence detector, holding an excitation wavelength of 366 nm and an emission wavelength of 455 nm, analyzed the dimedone products of pentadecanal and heptadecanal.

## 2. Results

### 2.1. Optimization of Fluorogenic Derivatization

The Hantzsch reaction is well characterized by the cyclization of two ß-diketones and aldehyde in the presence of ammonium acetate [19]. To determine the trace amount of long-chain fatty aldehydes pentadecanal and heptadecanal in biological samples, we employed two 5,5-dimethyl CHD molecules, and the reaction product with an aldehyde provided high sensitivity in the HPLC–fluorometric analysis. Figure 1 shows the 5,5-dimethyl CHD derivatization scheme and sampling for analysis.

In the first step, we investigated the 5,5-dimethyl CHD reaction conditions. Pentadecanal (100 pmol) dissolved in the S1PL reaction buffer (100 µL) was added to 1% 5,5-dimethyl CHD solution and then incubated to obtain the maximum fluorescent peak area, representing 5,5-dimethyl CHD reactivity. By a precise comparison of previously reported reaction conditions, pentadecenal also successfully derivatized in the customary reaction conditions with 1% 5,5-dimethyl CHD and yielded a high fluorescent peak area (Figure 2A). In the 1% 5,5-dimethyl CHD reaction condition, the optimal concentrations of acetic acid and ammonium acetate were 10% each (Figure 2B,C). The optimal reaction temperature was then chosen optimally at 70 °C, although fluorescence peaks were higher, in the range of 70–80 °C (Figure 2D). Finally, the optimized total reaction conditions were obtained with 10 mg/mL of 5,5-dimethyl CHD in H_2_O with 10% acetic acid and 10% ammonium acetate in the reaction medium at 70 °C and with a 60 min derivatization time (Figure 2E).

### 2.2. HPLC Separation of the S1PL Products

In HPLC conditions, an Agilent Eclipse XBD-C18 column and isocratic elution in a mixture of methanol and water containing 1% acetic acid (90:10, *v*/*v*) separated three peaks for the 5,5-dimethyl CHD derivatives of pentadecanal, hexadecanal, and the heptadecanal internal standard. The intensity of the fluorescence peaks at an excitation wavelength of 366 nm and an emission wavelength of 455 nm were stable during quantitative analysis. A typical chromatogram of the three derivatives was well separated and eluted in this condition’s carbon number order (Figure 3). However, pentadecenal from C17-S1P appeared faster than pentadecanal and overlapped with the extensive background peak eluted at around 4 min. The potential reaction byproducts were all eluted more quickly in 5 min after the 5,5-dimethyl CHD derivatives. The separated pentadecanal and heptadecanal symmetric peaks were eluted at 6.2 and 11.3 min, respectively.

For the analysis of lipid aldehydes produced in cell extracts, we investigated the chromatographic method validation. This HPLC method exhibited the sensitivity values in a limit of detection (LOD) of 0.8 pmol (0.38 ng) and a limit of quantitation (LOQ) of 2.5 pmol. This linearity was verified over a wide range from 1 to 1000 pmol (R^2^ = 0.9999) (Figure 4). Precision and accuracy were determined by spiking pmol levels of the standard aldehydes pentadecanal. Table 1 summarizes the precision and accuracy of pentadecanal spiked in F9-2 cell lysates.

### 2.3. Characterization of S1PL Activity in Mutant F9 Cell Lysates

We measured S1PL activity in S1PL-overexpressed F9-4 cell lysates with C17-Sa1P in 1% Triton X-100 and 570 μM of pyridoxal-5-phosphate (P5P) for 20 min. To optimize pentadecanal production, we measured S1PL activity for the indicated time (Figure 5). The S1PL activity exhibited linearity during the incubation from 0 to 30 min. However, the synthetic rate of pentadecanal gradually became lower, with a prolonged incubation of over 30 min. Therefore, incubation for 20 min was optimal in the quantitative range to measure pentadecanal production.

To understand the optimal cell lysate protein content for pentadecanal production, we then measured S1PL activity in the indicated protein amount. In high protein amounts over 50 μg, the pentadecanal production yield was slightly reduced but still useful for quantitative analysis (Figure 6). It is practically recommendable to use the linear relationship between S1PL and the protein content of cell lysates in the range of 0–50 μg (Figure 6).

We tested the S1PL affinity and pentadecanal production rate in a wide range of C17-Sa1P (1–160 μM) concentrations. In 50 μg of F9-4 cell lysate proteins and a 40 μM substrate concentration, pentadecanal production reached nearly maximal values. From the combination of the Michaelis–Menten equation plot (Figure 7A) and the Lineweaver–Burk transformation plot (Figure 7B), we calculated the S1PL enzyme kinetics in F9-4 cells: K_m_ 4.17 μM [3] and 182.43 pmol/mg protein/min (V_max_), respectively.

### 2.4. S1PL Activity in Three Mutant F9 Cells

To practically assess the applicability of S1PL activity in cell lysates, we measured S1PL activity, which converts C17-Sa1P into pentadecanal in three different F9 cells: F9-0 (wild type), F9-2 (S1PL^−/−^), and F9-4 (S1PL^+/+^) cells. The calculated S1PL activity was 24.5 (F9-0 cells), 3.6 (F9-2 cells), and 170.05 pmol/mg protein/min (F9-4 cells) (Figure 8A). 

For the evaluation of this method, FTY720 and 4-deoxypyridoxine (DOP), which are well-known S1PL enzyme inhibitors, were treated. We started the reaction with 50 μg of F9-4 cell lysates and 20 μM C17-Sa1P in the presence or absence of 50 μM FTY720 or 1000 μM DOP for 20 min. FTY720 (50 μM) inhibited 50% S1PL activity. At a relatively high DOP dose, a vitamin B6 antagonist reduced S1PL activity, suggesting that indirect vitamin B6 deficiency may influence S1PL activity. We also tested whether the deletion of the S1PL cofactor, vitamin B6 pyridoxal-5-phosphate (P5P), in the reaction conditions affects the S1PL assay in F9-4 cell lysates. Although the removal of P5P significantly reduced S1PL activity, there was a potential background of 25% S1PL activity in F9-4 (S1PL^+/+^) cells (Figure 8B). The results suggest that it is necessary to confirm the S1PL background level by deleting P5P when this method is applied to samples of biological origin.

## 3. Discussion

S1PL-mediated S1P cleavage produces (2E)-hexadecenal, and hexadecanal is the product of Sa1P degradation. These reactive fatty aldehydes can undergo further biotransformation to fatty acids or alcohols [20]. The abnormal expression of S1PL is associated with cancer development and developmental pathologies [21]. S1PL contributes to the control of lymphocyte egress from lymph nodes [22]. Therefore, the determination of S1PL activity is crucial in a variety of disease treatments. The monitoring of pentadecanal is not easily determined with MS spectrometry and fluorescence because a variety of bioactive aldehydes are highly reactive, which may produce multiple unspecified compounds.

The specific derivatization of pentadecanal with two molecules of 5,5-dimethyl CHD was highly selective to fatty aldehydes, which readily formed a new fluorophore product with a 455 nm emission wavelength suitable for the HPLC–FLD system (Figure 1). This method’s advantage is the specific conversion of pentadecenal with a nonfluorescent reagent to a strong fluorescent product (Figure 3). This fluorescent S1PL assay’s detection limit was almost 0.38 ng (0.8 pmol), giving higher sensitivity in the measurement of lipid aldehydes. The second advantage is the use of F9-4 cell lines, which S1PL stably overexpressed, providing sevenfold higher S1PL activity. The distinct characteristic step of C18 solid-phase extraction specifically traps the final reaction fluorescent product efficiently, showing a clear background chromatogram in the region of fatty aldehydes eluted from cell extracts (Figure 3C). The optimized derivatization conditions ensured the reliable detection of pentadecenal as low as 0.8 pmol, sufficient to monitor S1PL activity in culture cells.

The linear regression (R^2^ = 0.9999) represents wide linearity to 1000 pmol of standard aldehydes. However, we considered the oxidation or reduction of pentadecanal to 2-pentadecen-1-ol or 2-pentadecenoic acid. In practice, the longer incubation time for S1PL activity of over 30 min exhibited a reduction in pentadecanal production (Figure 5). Furthermore, as shown in Figure 6, the protein amount also influenced pentadecanal formation. The S1PL reactivity of pentadecanal formation is dependent on the concentration of the protein amount and reaction time for the quantitative assay. We obtained the best result in the conditions of 50 µg of cell lysate proteins with a 20 min reaction time.

We then measured S1PL activity and enzyme kinetics in F9-4 (S1PL^+/+^) cells by employing 50 µg protein per reaction, a 20 min reaction time, and various C17-Sa1P concentrations. The 4.17 µM K_m_ values obtained were similar to the previous K_m_ values of 9.0–20.1 µM with Sa1P as the substrate [14]. The stably overexpressed S1PL F9-4 cells and dominant-negative F9-2 cells showed an almost similar morphological shape and proliferation rate than the normal F9-0 cells (data not shown). We also reconfirmed the sevenfold higher S1PL activity in F9-4 cells compared to wild type F9-0 cells (Figure 8A). In this system, the tested S1PL inhibitors, FTY720, and DOP efficiently reduced pentadecanal production in F9-4 cells. We found that S1PL activity remained when the S1PL cofactor, P5P, was not added to the reaction mixture (Figure 8B). The S1PL enzyme might use the residual intracellular P5P for the S1PL reaction in cell lysates. Therefore, it is necessary to check the background S1PL activity in cell lysates.

## 4. Materials and Methods

### 4.1. Materials

Methanol, water, and acetonitrile (HPLC grade) were purchased from Fisher (Pittsburgh, PA, USA). S1P, C17 Sa1P, and hexadecenal were obtained from Avanti Polar Lipids (Alabaster, AL, USA). Pentadecanal and the heptadecanal internal standard were obtained from Tokyo Chemical Industry (Tokyo, Japan). 5,5-Dimethyl cyclohexanedione was obtained from Sigma (St. Louis, MO, USA). FTY720 and 2-acetyl-5-tetrahydroxybutyl imidazole (THI) were purchased from Enzo Life Sciences (New York, NY, USA). Solid-phase extraction cartridges were obtained from Waters (Massachusetts, MA, USA). All other chemicals were obtained from Sigma (St. Louis, MO, USA).

### 4.2. Cell Culture and Preparation for S1PL Assay

Dr. A. Kihara kindly supplied mouse embryonal carcinoma F9-0 cells and F9-2 cells (S1PL knocked-out) as well as F9-4 cells (S1PL overexpressed) (Hokkaido University, Japan). Cells were grown in Dulbecco’s modified Eagle’s medium (DMEM) containing 10% (*v*/*v*) FBS and 1% penicillin–streptomycin in 0.1% gelatin-coated dishes. Cells were cultured at 37 °C in a humidified 5% CO_2_ atmosphere and routinely subcultured every other day using a solution of trypsin–EDTA from Life Technologies, Inc. (Gaithersburg, MD, USA). Cells were washed twice with ice-cold PBS and then scraped in 1 mL of S1PL extraction buffer (5 mM Mops, 1.0 mM EDTA, 0.25 M sucrose, 1.0 mM PMSF), Then, 1× protease inhibitor and 10% (*v*/*v*) glycerol at pH 7.4 were sonicated for 10 s. We removed cell debris via low-speed centrifugation (1000× *g* for 5 min), and the protein concentration of the supernatant was determined by using the Pierce bicinchoninic acid (BCA) reagent (Thermo Fisher Scientific, Rockford, IL, USA)

### 4.3. Preparation of the Dimedone Reagent

We prepared the stock solution consisting of 10 mL of acetic acid and 10 g of ammonium acetate in 100 mL of water. Then, 1 g of dimedone was added to the solution and shaken to prepare the reagent solution. We stored the derivatization reagent at 4 °C in the dark.

### 4.4. S1PL Assay

The S1PL reaction was initiated by mixing 10 μL of 400 μM C17 Sa1P in 1% Triton X-100 in water, 140 μL of reaction buffer (35 mM potassium phosphate buffer (pH 7.4)), 0.6 mM EDTA, 70 mM sucrose, 36 mM sodium fluoride, 570 μM pyridoxal-5′-phosphate (P5P), and 50 μL of protein preparation in lysis buffer (50 μg of protein) in Eppendorf tubes. The reaction was performed for 20 min at 37 °C and was stopped with the addition of 200 μL of ethanol. We added the heptadecanal internal standard (100 pmol) and 200 μL of 5,5-dimethyl CHD reagent, and an aldehyde-5,5-dimethyl CHD reaction occurred directly in the heated block (75 °C) for 60 min.

We performed solid-phase extraction (SPE) using Sep-Pak C18 cartridges (100 mg). The columns were preconditioned with 2 mL of methanol and then 2 mL of water. After loading the mixture (600 μL) on the SPE column, we washed the column with 2 mL of MeOH–H_2_O (5:5, *v*/*v*) and eluted the derivatized aldehydes with 1 mL of methanol. We dried the collected methanol fraction in a speed vacuum system (Vision, Seoul, Korea).

### 4.5. Measurement of Aldehydes

We dissolved the residues in 200 μL of 0.1% acetic acid in methanol. For the determination of pentadecanal and heptadecanal internal standard, we used a 1260 Infinity liquid chromatography from Agilent Technologies (Waldbronn, Germany) combined with a fluorescence detector. We separated pentadecanal and heptadecanal derivatives by using an Eclipse XDB-C18 column (4.6 mm × 150 mm, 5.0 μm particle size) with elution (MeOH/H_2_O/acetic acid, 90:10:0.1, *v*/*v*) for 20 min at a flow rate of 2 mL/min at the fluorescence detection (λex = 366 nm, λem = 455 nm) of the labeled analytes.

### 4.6. Statistics

All values were expressed as a mean ± standard deviation (SD). Statistical significance was assessed with one-way analysis of variance (ANOVA) with the Newman–Keuls test, and in all cases, the criterion for significance was *p* < 0.01.

## 5. Conclusions

In conclusion, we developed a new S1PL activity assay with 5,5-dimethyl CHD derivatization, providing a low background in the HPLC detection system and enabling the measurement of S1PL activity.

## Figures and Tables

**Figure 1 ijms-22-01438-f001:**
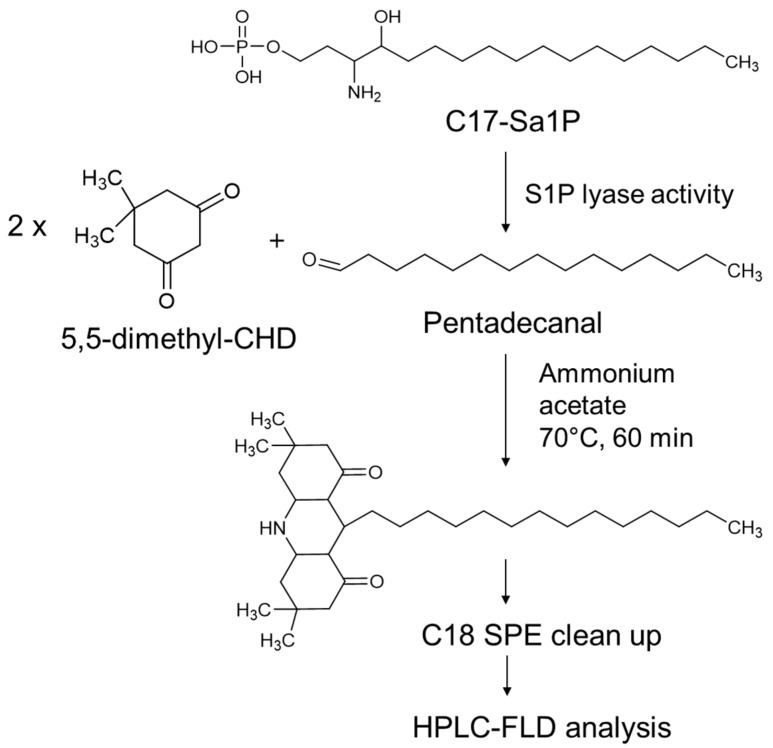
Scheme of sphingosine-1-phosphate lyase (S1PL) activity assay using HPLC–FLD analysis.

**Figure 2 ijms-22-01438-f002:**
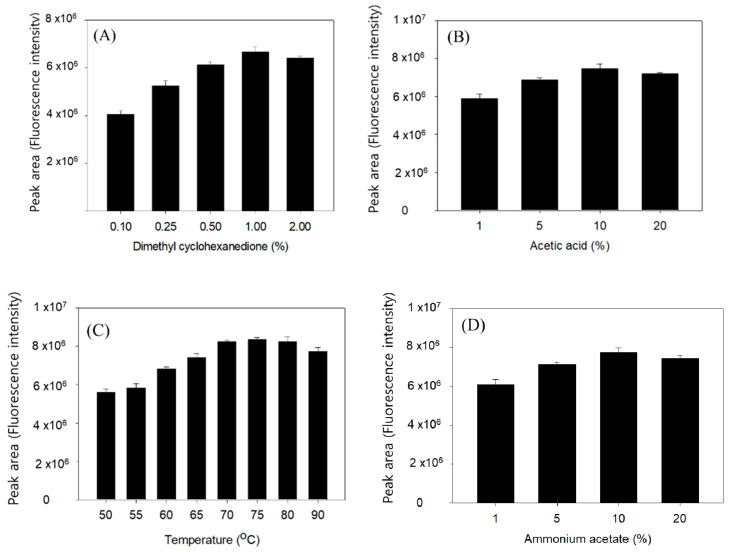
Optimal conditions for the 5,5-dimethyl cyclohexanedione (5-dimethyl CHD) derivatization of pentadecanal and the effect of derivatization. (**A**) 5,5-Dimethyl CHD concentration, (**B**) acetic acid concentration, (**C**) ammonium acetate concentration, (**D**) temperature, and (**E**) reaction time for the formation of pentadecanal derivatives. Pentadecanal (100 pmol) spiked into the S1PL reaction buffer and derivatized with 5,5-dimethyl CHD at various conditions. After solid-phase extraction, we analyzed the lipid aldehyde content in the dried samples. Each point represents the mean ± SD of triplicate analyses.

**Figure 3 ijms-22-01438-f003:**
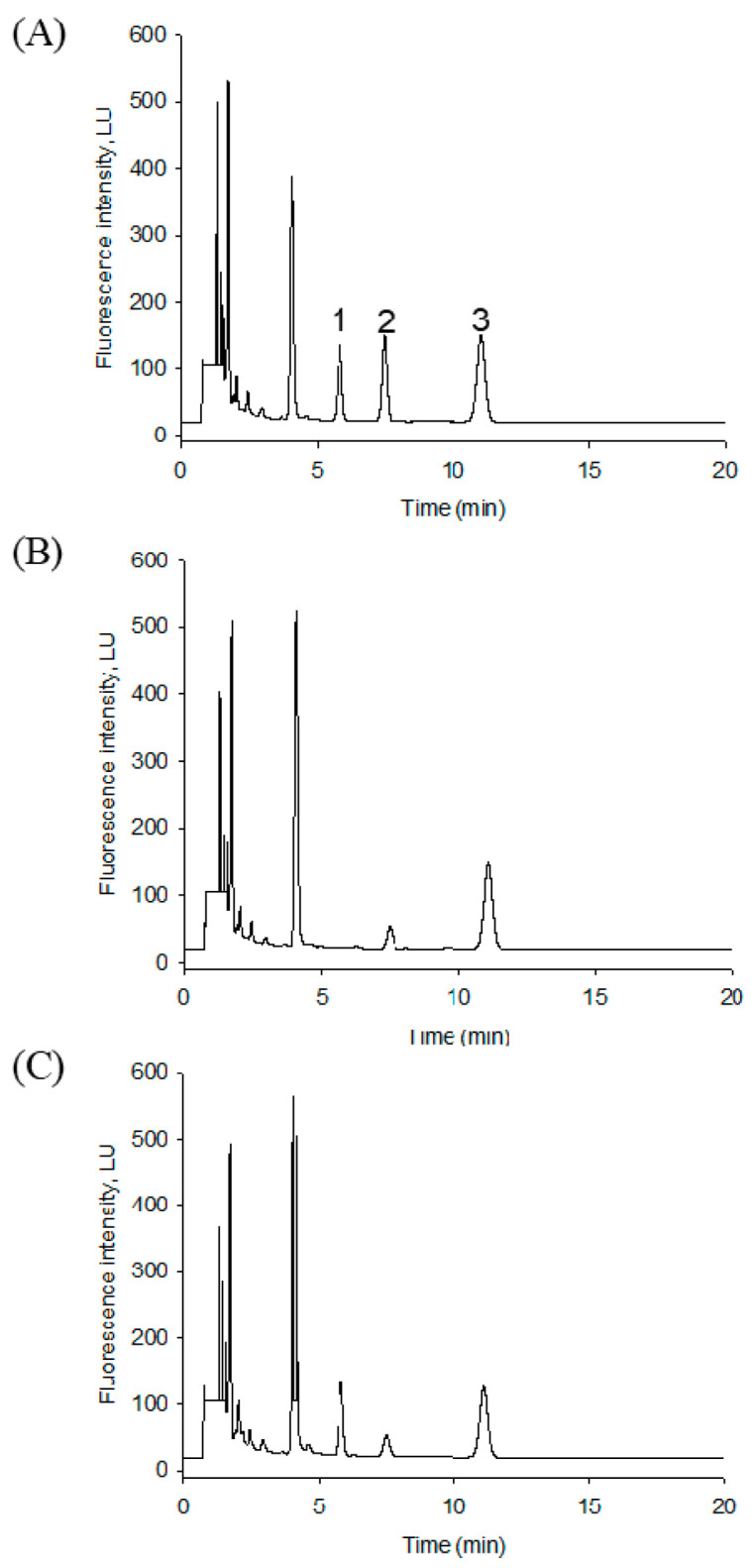
Typical HPLC–FLD chromatograms. Profile of 5,5-dimethyl CHD derivatives of (**A**) 100 pmol of pentadecanal, hexadecanal, and the heptadecanal standard solution. (**B**) Blank cell lyases with 100 pmol heptadecanal. (**C**) F9-4 cell S1PL activity after solid-phase extraction (SPE) and C18 reverse-phase HPLC–FLD with 90% methanol and 0.1% acetic acid as the elution. The SPE extraction yield was above 85%. 1: pentadecanal; 2: hexadecanal; 3: heptadecanal.

**Figure 4 ijms-22-01438-f004:**
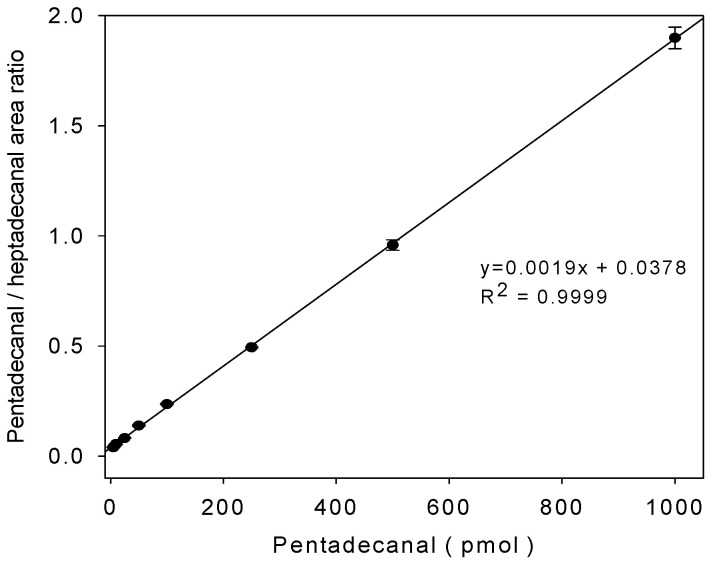
Calibration curve for the quantitation of pentadecanal. We added the indicated concentrations of the pentadecanal standard to the cell lysates. After the 5,5-dimethyl CHD derivative, we analyzed the SPE extract using C18 reverse-phase HPLC. The fluorescent peak area ratio of pentadecanal versus heptadecanal showed a linear curve with increasing pentadecanal concentrations. The data represent the mean ± SD of triplicate analyses.

**Figure 5 ijms-22-01438-f005:**
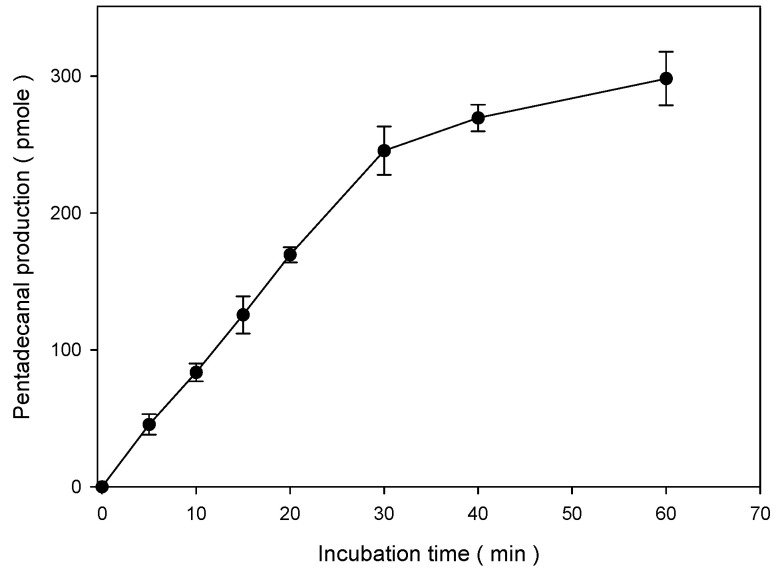
Time course of pentadecanal production in F9-4 cell lysate proteins. S1PL was measured with 50 μg of F9-4 cell lysate proteins and 20 μM C17-Sa1P for the indicated time. We stopped the 5,5-dimethyl CHD reaction by adding 200 μL of ethanol. Lipid aldehyde derivatives were extracted via solid-phase extraction and analyzed using C18 reverse-phase HPLC. The data represent the mean ± SD of triplicate analyses.

**Figure 6 ijms-22-01438-f006:**
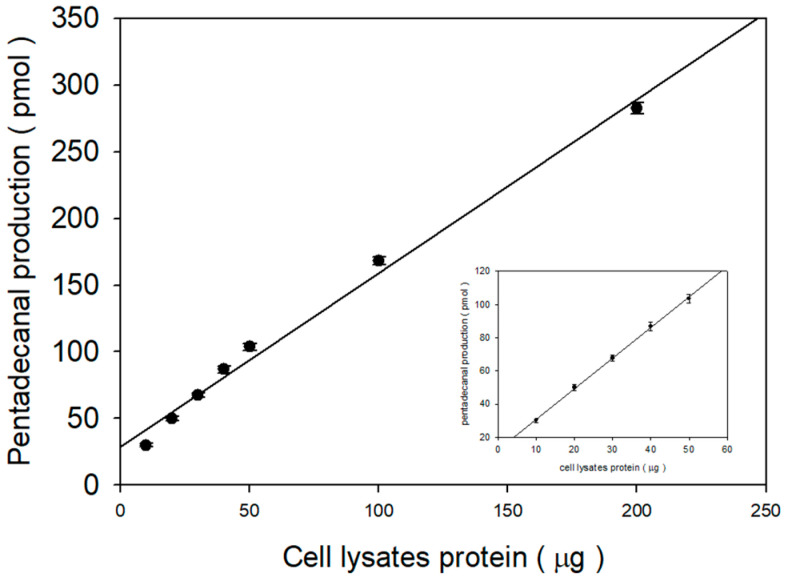
Linearity of the S1PL reaction at higher protein concentrations after applying a correction for biotransformation. We performed the response for 20 min with 20 μM C17 Sa1P as the substrate and variable amounts of F9-4 cell lysate proteins. The 200 μL volume of ethanol stopped the pentadecanal production reaction. For the derivatization reaction, we added the 5,5-dimethyl CHD solution with 100 pmol of heptadecanal as the internal standard. Aldehyde-5,5-dimethyl CHD was extracted via solid-phase extraction and analyzed using C18 reverse-phase HPLC–FLD. The insert shows expanded views of the data generated at high cell lysate protein concentrations. The data represent the mean ± SD of triplicate analyses.

**Figure 7 ijms-22-01438-f007:**
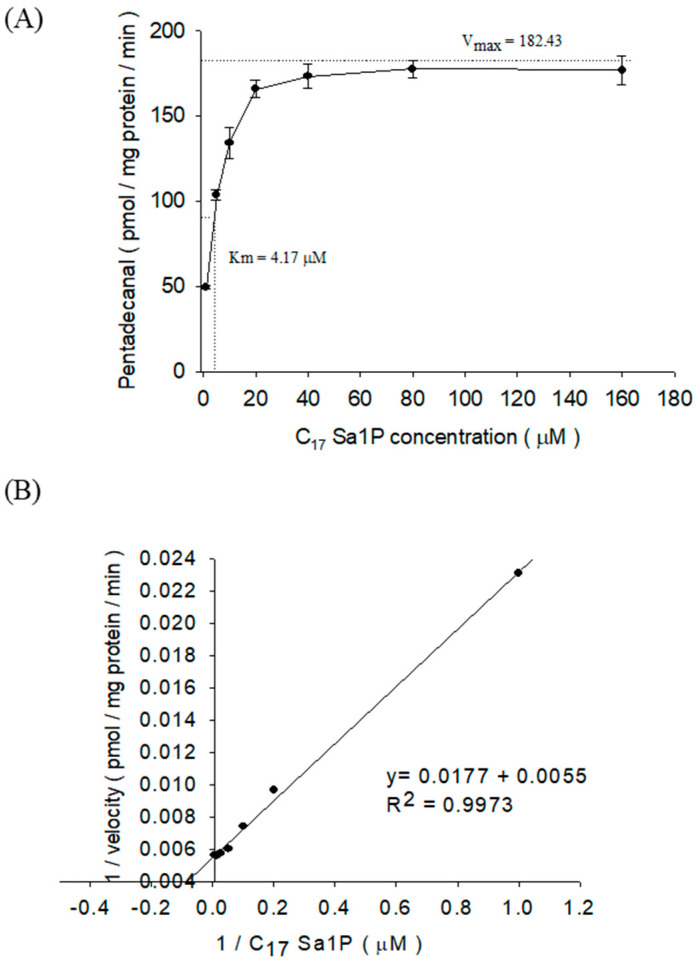
S1PL activity as a function of substrate concentration (**A**) and the S1PL reaction for 20 min with variable amounts of C17 Sa1P and 50 μg of cell lysate proteins (**B**). The data represent the mean ± SD of triplicate analyses.

**Figure 8 ijms-22-01438-f008:**
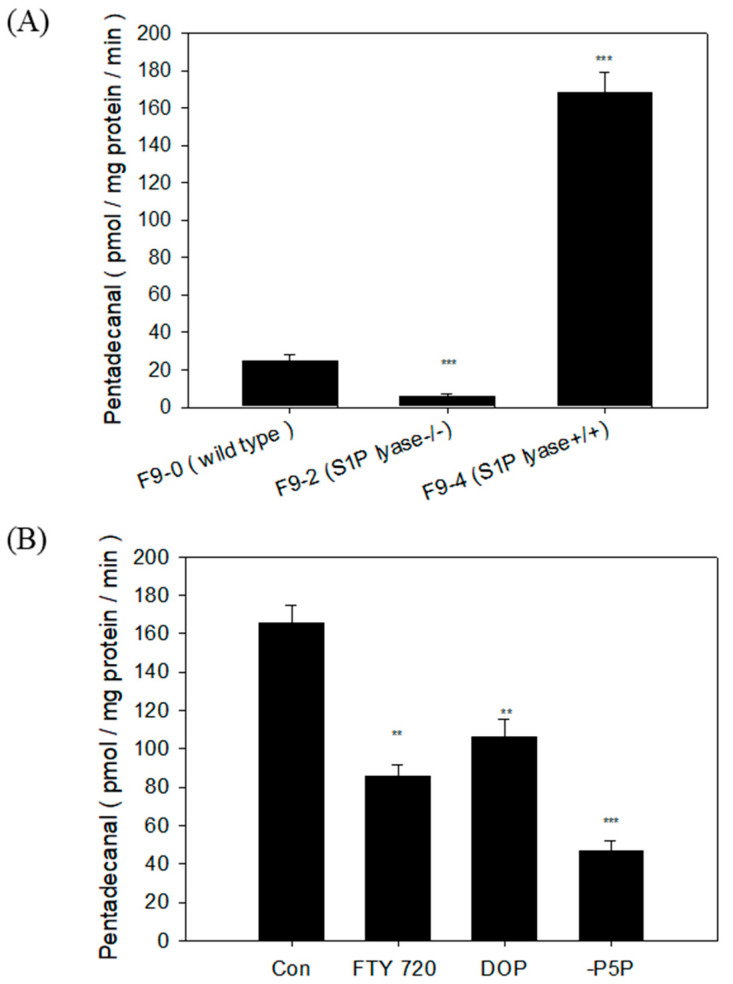
S1PL activity assay in the S1PL modification cell line. FTY720 and 4-deoxypyridoxine (DOP) inhibited S1PL activity in vitro. (**A**) S1PL reaction for 20 min with 20 μM C17-Sa1P as the substrate and 50 μg of F9-0 (wild type), F9-2 (S1PL^−/−^), and F9-4 (S1PL^+/+^) cell lysates in the presence or absence of 0.57 mM pyridoxal-5-phosphate (P5P). (**B**) S1PL reaction in the presence or absence of 0.57 mM P5P with FTY720 (50 μM) or DOP (1000 μM). The data represent the mean ± SD of triplicate analyses. ** *p* < 0.01 and *** *p* < 0.001 compared to the F9-0 cell or control groups.

**Table 1 ijms-22-01438-t001:** Precision and Accuracy of Pentadecanal in F9-2 Cell Lysates.

Component	Spike Amount (pmol)	Intraday (RSD, %)	Interday (RSD, %)	Intraday (Accuracy, %)	Interday (Accuracy, %)
	2.5	7.23	9.12	93.60 ± 6.75	91.92 ± 8.21
	5	3.25	3.18	95.29 ± 3.08	96.07 ± 3.28
Pentadecanal					
	50	2.34	2.47	103.41 ± 3.08	102.74 ± 2.67
	400	1.64	1.78	98.54 ± 1.96	100.06 ± 2.78

Accuracy (%) = calculated concentration/theoretical concentration × 100, RSD (%) = standard deviation of the concentration/mean concentration × 100.

## Data Availability

Not applicable.

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
