# Peer review of "A Bioassay Using a Pentadecanal Derivative to Measure S1P Lyase Activity"

_ijms, 2021, doi:10.3390/ijms22031438_

Round 1

Reviewer 1 Report

This manuscript about ‘a new assay for 1p lyase enzyme’ is well documented. Previously, the substrates for S1P lyase used were isotope- or fluorophore-tagged substrates. Therefore, S1P lyase activities were not to say it specific assay, depending on the substrate characters. On this point, I think that C17-S1P, one carbon deleted from S1P will be a good substrate. During reading the manuscript, I raised up some questions which are not clearly understood and thus demand the explanation more clearly. This manuscript will be publishable as a new assay introduction to IJMS.

  1. S1P lyase digest S1P into hexadecenal and phosphoethanolamine as explained in introduction. Instead, C17-Sa1P was used to product pentadecanal which lacks a double bond. Why you did not use C17-S1P as an optimal substrate?
  2. For obtaining more sensitive analysis for lipids, LC-MS/MS system was demanded to measure in biological samples. I would recommend to authors to discuss more about the merit of this method.
  3. SPE protocol is not clear. 5,5-dimethyl CHD labeled aldehydes were washed and eluted. How much internal standard you added? How much the absolute yield obtained from SPE sample pretreatment?
  4. What is DHS1P? Is it different to Sa1P?
  5. How about the solubility of So-1-P up to 50 uM? Is there no problem without any specific procedure for dissolving?
  6. Misspelling: lane 271 - 5% CO2 atmosphere. Please check typos

Author Response

This manuscript about ‘a new assay for 1p lyase enzyme’ is well documented. Previously, the substrates for S1P lyase used were isotope- or fluorophore-tagged substrates. Therefore, S1P lyase activities were not to say it specific assay, depending on the substrate characters. On this point, I think that C17-S1P, one carbon deleted from S1P will be a good substrate. During reading the manuscript, I raised up some questions which are not clearly understood and thus demand the explanation more clearly. This manuscript will be publishable as a new assay introduction to IJMS.

S1P lyase digest S1P into hexadecenal and phosphoethanolamine as explained in introduction. Instead, C17-Sa1P was used to product pentadecanal which lacks a double bond. Why you did not use C17-S1P as an optimal substrate?

  • By S1P lyase digestion, C17-S1P is converted pentadecenal and phosphoethanolamine. Unfortunately, pentadecenal was eluted faster than pentadecanal and nearly overlap with the large background peak eluted around 4 minutes in Figure 3.
  • The substrate specificity of S1P lyase on Sa1P was comparable to S1P.
  • We inserted the detailed sentence in 117 lane “However, the pentadecenal from C17-S1P appeared faster than pentadecanal was overlap with the extensive background peak eluted around 4 minutes. ”

For obtaining more sensitive analysis for lipids, LC-MS/MS system was demanded to measure in biological samples. I would recommend to authors to discuss more about the merit of this method.

  • We already described the merit of this method in Discussion section as lane 223 “This method's advantage is a specific conversion of pentadecenal with a non-fluorescent reagent to a strong fluorescent product (Figure 3).”
  • We added more in lane 225 “This fluorescent S1PL assay's detection limit was almost 0.38 ng (0.8 pmol), giving higher sensitivity on lipid aldehydes measurement. The second advantage is to use the F9-4 cell lines, which stably S1PL overexpressed that provide 7-fold higher S1PL activity.”

SPE protocol is not clear. 5,5-dimethyl CHD labeled aldehydes were washed and eluted. How much internal standard you added? How much the absolute yield obtained from SPE sample pretreatment?

  • Internal standard concentration is already described in lane 296 : heptadecanal (100 pmol) after S1PL reaction.
  • We already described SPE protocol in 4.4 S1PL assay (lane 276).
  • However, for precise protocol, we modified the protocol in detail.
  • We described in lane 284 as “ We performed solid-phase extraction (SPE) using Sep-Pak C18 cartridges (100 mg). The columns were preconditioned with 2 mL methanol and then 2 mL water. After loading the mixture (600 μL) on the SPE column, we washed the column with 2 ml of MeOH–H2O (5:5, v/v), and the derivatized aldehydes eluted with 1 mL methanol. We dried the collected methanol fraction in a speed vacuum system (Vision, Seoul, Korea).”
  • The absolute SPE yield is above 85%. As you see the peak 3 in Figure 3(B) compared to the peak 3 in Figure 3(A). However, to reduce the quantification error, we spiked internal standard, heptadecanal for data normalization. We insert in Figure 3 legend as “The SPE extraction yield was above 85%, respectively.”

What is DHS1P? Is it different to Sa1P?

  • We are sorry to make you mislead. Dihydrosphingosine 1-phosphate (DHS1P) is the same molecules of sphinganine 1-phosphate (Sa1P). We changed DHS1P into Sa1P.

How about the solubility of So-1-P up to 50 uM? Is there no problem without any specific procedure for dissolving?

  • The solubility of C17 Sa1P is not good in organic solvents. We described in lane 277 as We prepared “400 uM C17 Sa1P stock solution with 1% Triton X-100 in water”. Before doing S1PL assay, stock solution was homogenized by warming the tube for 15 minutes at 37 oC heating block shaker.
  • In this condition, we do not meet any problem for the reproducibility.

Misspelling: lane 271 - 5% CO2 atmosphere. Please check typos

  • We checked and corrected all the sentences by Grammarly PREMIUM.

Reviewer 2 Report

The manuscript ´S1P lyase assay to measure pentadecenal by 5,5-dimethylcyclohexanedione derivatization´ submitted by Shin et al. describes a method for measurement of S1P-lyase activity after derivatization of pentadecanal with 5,5-dimethyl CHD leading to a fluorescent reaction product, which can be determined via fluorescence detection.

There are a variety of concerns that should be addressed:

Figure 1 show the schematic derivatisation mechanism. The structures should be presented with an adequate program.

In Figure 3 the 3 different aldehydes are presented. Fig. 3B shown blank cell lysate (instead of lyases). But it seems that the internal standard heptadecanal was added.

The authors indicated that high protein amounts tends to slightly reduce formation due to biotransformation. What kind of biotransformation will happen to pentadecanal, is there an oxidation to the fatty acid and Co-A activation. This can be proofed via inhibitors. 

In the introduction section the authors state that strong mass ions originated from reagent itself sometimes give high background in TIC chromatograms. This plays no role when using the MRM modus.

The title makes no sense, it is an S1P lyase assay using pentadecanal and its derivatization to measure activity. 

The number of spelling errors is extremely high.

Author Response

The manuscript ´S1P lyase assay to measure pentadecenal by 5,5-dimethylcyclohexanedione derivatization´ submitted by Shin et al. describes a method for measurement of S1P-lyase activity after derivatization of pentadecanal with 5,5-dimethyl CHD leading to a fluorescent reaction product, which can be determined via fluorescence detection.

There are a variety of concerns that should be addressed:

Figure 1 show the schematic derivatisation mechanism. The structures should be presented with an adequate program.

  • Yes, we re-designed and make it to understand more easily the derivatization scheme by ChemDraw program.

In Figure 3 the 3 different aldehydes are presented. Fig. 3B shown blank cell lysate (instead of lyases). But it seems that the internal standard heptadecanal was added.

-          Your comment is exactly right. 100 pmol heptadecanal was spiked to cell lysates because it is used internal standard for pentadecanal quantification. As you see (B) chromatogram, endogenous hexadecanal also be detected. Therefore, we can observe S1P lyase activity of the conversion from C17-Sa1P into pentadecanal without any disturbance.

-          . Thus, the legend in Figure 3 is changed clearly as below.

-          and (B) blank cell lyases with 100 pmol heptadecanal.

The authors indicated that high protein amounts tends to slightly reduce formation due to biotransformation. What kind of biotransformation will happen to pentadecanal, is there an oxidation to the fatty acid and Co-A activation. This can be proofed via inhibitors.

  • Your question is correct.
  • Many references including we cited suggested the diversity of mammalian resources of fatty aldehydes specific enzymes, peroxisomal fatty acyl-CoA reductase 1(1), aldehyde dehydrogenase ALDH3B1(2) and microsomal aldehyde reductase(3) are possibly involved to degrade pentadecanal in mouse embryonal carcinoma cell lines (F9-4) .
  • This biotransformation will be proved by specific inhibitors, which may block pentadecanal degradation and thus enhanced the assay sensitivity. However, we do not guarantee whether the use of specific inhibitors can specifically block pentadecanal biotransformation because of diverse enzymes also can metabolize pentadecanal. Furthermore, S1p lyase enzyme may be influenced by treated inhibitors.
  • To escape practically this deviation in high protein amounts, we therefore used 50 µg of cell lysates protein that was maximal dose to exhibit the linearity on S1P lyase activity (Figure 6, insert). In this condition, we successfully detect pentadecanal amount by high-sensitive fluorescent derivatization. We explained already on lane 158 – 162.
  • An alternative membrane topology permits lipid droplet localization of peroxisomal fatty acyl-CoA reductase 1.

Exner T, Romero-Brey I, Yifrach E, Rivera-Monroy J, Schrul B, Zouboulis CC, Stremmel W, Honsho M, Bartenschlager R, Zalckvar E, Poppelreuther M, Füllekrug J

J Cell Sci. 2019 Mar 18;132(6):jcs223016. doi: 10.1242/jcs.223016.

  • Substrate specificity, plasma membrane localization, and lipid modification of the aldehyde dehydrogenase ALDH3B1.

Kitamura T, Naganuma T, Abe K, Nakahara K, Ohno Y, Kihara A.

Biochim Biophys Acta. 2013 Aug;1831(8):1395-401. doi: 10.1016/j.bbalip.2013.05.007. Epub 2013 May 27.

  • Participation of microsomal aldehyde reductase in long-chain fatty alcohol synthesis in the rat brain.

Takahashi N, Saito T, Goda Y, Tomita K.

Biochim Biophys Acta. 1988 Nov 25;963(2):243-7. doi: 10.1016/0005-2760(88)90287-1.

 In the introduction section the authors state that strong mass ions originated from reagent itself sometimes give high background in TIC chromatograms. This plays no role when using the MRM modus.

  • Thank you so much for your comments. We changed the sentence as shown below in lane 66.

“However, the intense mass fragment ions originated from the reagent itself provide high sensitivity in MRM mode, non-specific reaction with other functional groups in biological samples sometimes exhibit complex chromatogram, which disturbs the detection of corresponding aldehydes. Therefore, the ideal designed reagent is that the non-fluorescence and reacts with aldehyde selectively form the fluorogenic structure.”

 The title makes no sense, it is an S1P lyase assay using pentadecanal and its derivatization to measure activity.

  • We follow reviewer’s comment on the title.
  • We changed the title “A bioassay using pentadecanal derivative to measure S1P lyase activity”

The number of spelling errors is extremely high.

  • We checked and corrected all the sentences by Grammarly PREMIUM.
  • We retried the spell check and amended to whole manuscript to be acceptable to IJMS

Round 2

Reviewer 2 Report

The authors addressed all of the concerns. However, the quality of figure 1 is not sufficient. The English language still needs some editing.

Author Response

Reviewer 2

We are happy to hear the valuable words for polishing the manuscript and hope to be satisfied with additional modifications.

Reviewer 2 commented as below.

"The authors addressed all of the concerns. However, the quality of figure 1 is not sufficient. The English language still needs some editing."

  • In Figure 1 modification, we enlarged the letters and the size of the C17-Sa1P chemical structure. Figure 1 could summarize the whole assay process at once.
  • For Figure 1, we already follow Referee 2's comment and modified Figure 1 for summarizing S1PL assay to produce hexadecanal from C17-Sa1P, The newly created hexadecanal is derivatized by two molecules of 5,5-dimethyl-CHD for sensitive fluorescence detection.
  •  

For the second comment: The English language still needs some editing."

  • We checked English grammar pass the Grammarly PREMIUM system in the first revision.
  • We cleared most of the explanation of the crucial results.
  • For reference, we copied and attached Grammarly PREMIUM's ability and the payment for use it.
